# Self-Myofascial Release of the Foot Plantar Surface: The Effects of a Single Exercise Session on the Posterior Muscular Chain Flexibility after One Hour

**DOI:** 10.3390/ijerph20020974

**Published:** 2023-01-05

**Authors:** Luca Russo, Eleonora Montagnani, Davide Pietrantuono, Fabiola D’Angona, Tommaso Fratini, Riccardo Di Giminiani, Stefano Palermi, Francesco Ceccarini, Gian Mario Migliaccio, Elena Lupu, Johnny Padulo

**Affiliations:** 1Department of Human Sciences, Università Telematica degli Studi IUL, 50122 Florence, Italy; 2Department of Sports and Health Sciences, University of Brighton, Brighton BN2 4AT, UK; 3Department of Medicine and Health Sciences, University of Molise, 86100 Campobasso, Italy; 4Department of Biotechnological and Applied Clinical Sciences, University of L’Aquila, 67100 L’Aquila, Italy; 5Public Health Department, University of Naples Federico II, 80132 Naples, Italy; 6Department of Psychology, New York University Abu Dhabi, Abu Dhabi 129188, United Arab Emirates; 7Department of Performance, Sport Science Lab, 09131 Cagliari, Italy; 8Department of Motor Activities, Petroleum Gas University Ploiesti, 100600 Ploiesti, Romania; 9Department of Biomedical Sciences for Health, Università degli Studi di Milano, 20133 Milan, Italy

**Keywords:** self-myofascial release, training, sit and reach, posterior muscular chain, flexibility

## Abstract

This study evaluated the effects of a single exercise session of Self-Myofascial Release (SMR) on the posterior muscular chain flexibility after one hour from the intervention. Thirty-six participants performed SMR using a rigid ball under the surface of both feet. Participants were tested with the Sit and Reach (S&R) test at four different times: before (T0), immediately after (T1), 30 (T2), and 60 (T3) minutes after the SMR intervention. The sample (*n* = 36) was categorized into three groups: (1) flexible, (2) average, and (3) stiff, based on the flexibility level at T0 (S&R values of >10 cm, >0 but <10 cm and <0 cm, respectively). For the whole sample, we detected significant improvements in the S&R test between the T1, T2, and T3 compared to T0. The stiff group showed a significant (*p* < 0.05) improvement between T1–T2 and T1–T3. Results were similar between the average group and the whole sample. The flexible group did not show any significant difference (*p* > 0.05) over time. In conclusion, this investigation demonstrated that an SMR session of both feet was able to increase posterior muscular chain flexibility up to one hour after intervention. Considering that a standard training session generally lasts one hour, our study can help professionals take advantage of SMR effects for the entire training period. Furthermore, our results also demonstrate that physical exercise practitioners should also assess individuals’ flexibility before training, as the SMR procedure used in this work does not seem necessary in flexible individuals.

## 1. Introduction

Self-Myofascial Release (SMR) is a common self-treatment strategy used by a wide plethora of physically active people, such as gym or home fitness users [1,2], high-level athletes [3,4,5], and global health and postural trainers [6,7,8]. According to the literature, SMR can be defined as a subcategory of myofascial release [4,9,10], which is a series of manipulative techniques where pressure is applied to soft tissue, with regard to muscles and fascia [11]. In fact, the main declared goal of SMR is to mobilize the targeted soft tissues with rolling devices, such as foam rollers or hard balls, which are used by applying a certain amount of pressure over the surface of the skin, thus creating a self-induced massage effect. The effectiveness of SMR is well explained in the literature and different authors adduce several mechanisms of action. Compression may increase fascial elasticity thanks to a temporary change in water content [12]. After the compression, the local blood flow is increased, facilitating the removal of metabolites and delivering oxygen [13,14,15] as well as the warming of the tissue, ripping the restrictions within layers of the fascia and restoring soft tissue elasticity [16], reducing the inflammation status [9]. The pressure on soft tissue also leads to a mechanoreceptor stimulation, which is then able to reduce muscular and fascia tension [17]. All these mechanisms are well-known and described in detail in previously published reviews [9,10]. Although some authors are still dubious about the proper name to use for these techniques applied on the myofascial tissue [18], a growing amount of evidence suggests positive outcomes of SMR, such as decreasing pain, reduction of delay onset muscle soreness (DOMS), faster physical recovery, and an increase in muscle flexibility and joint range of motion (ROM) without a reduction in force production [9,19,20].

Doubtless, the increase of joint ROM and muscle flexibility are two of the most interesting aspects for coaches and sport science practitioners. Flexibility is a key factor for several sports [21] as well as for daily life tasks and musculoskeletal health [22,23,24]. Stretching is classically considered the best way to improve flexibility over time [25], but evidence suggests both chronic and acute positive effects of SMR on flexibility [9,19,26,27], despite some doubts remaining [28].

The acute effects of SMR on flexibility are the most investigated because of the widespread use of SMR techniques before and after physical training. In fact, it is very common to see athletes and fitness users adopt foam rollers or rubber balls before training, especially on the lower limbs and the back. In addition, the literature gives great attention to the effects of SMR, especially on the posterior muscular chain [29,30,31]. The posterior muscular chain encompasses a series of muscles interlinked by the deep fascia, extending from the foot to the fascial sheath of the eyeball [32]. For this reason, the myofascial chains are usually related to the whole-body posture [33,34]. The principal acute effects of SMR on the posterior muscular chain are (1) increasing flexibility of the whole chain or specific muscles (e.g., hamstrings) [28,29] and (2) increasing ankle and spine ROM [29,31]. It must be underlined that these effects are evoked by performing SMR on any segment of the posterior muscular chain [31].

Although the acute effects of SMR on the posterior muscular chain are supported by strong evidence, very few data are available on the duration of this “lengthening” effect. Researchers have shown how the SMR effects last up to 10 min [16,35], and other data confirmed the duration of such effects for up to 30 min [36]. However, the longer-term effects of SMR on posterior muscular chain flexibility have not been demonstrated yet. Moreover, no data are available regarding the acute duration of SMR in people with different levels of posterior chain flexibility. This information could be very useful in terms of practice and training, to properly organize the exercises along the training sessions and use SMR stimulation with a specific timing before the training session or competition.

To our knowledge, no studies have investigated the duration of the acute effects of an SMR on the feet plantar muscles in different clusters of people according to their flexibility level. In fact, it is reasonable to speculate that SMR stimulation offers higher ROM gain in individuals with a lack of posterior muscular chain flexibility. Therefore, the aims of this investigation were twofold: (1) measure the flexibility obtained during a one-hour span after the SMR procedure, and (2) measure the flexibility obtained in subgroups of individuals divided by their flexibility levels, which will be defined before the SMR procedure.

## 2. Materials and Methods

### 2.1. Participants

Thirty-six volunteer individuals (age 23.9 ± 1.9 years; body height 171.7 ± 10.8 cm; body mass 68.4 ± 15.1 kg) were selected for the study. An a priori analysis was conducted for the sample size setting the error probability at 5% and the power of the analysis at 80%. The recommended sample was 24 participants, but we eventually recruited 36 individuals for the entire sample. Moreover, another a priori analysis was conducted for the sub-groups setting the effect size at 0.4, the error probability at 5%, and the power of the analysis at 80%, obtaining 10 participants for the group. The entire sample was divided into 17 males (age 24.2 ± 1.3 years; body height 180.2 ± 8.3 cm; body mass 80.5 ± 12.9 kg) and 19 females (age 23.7 ± 2.3 years; body height 164.1 ± 5.9 cm; body mass 57.5 ± 5.7 kg). Participants were included in the study if they: (1) reported to be in good health and physically active (declaring to perform at least 150 min/w of physical activity); (2) did not suffer any accidents or injuries in the 12 months preceding the test; (3) did not report pain in the ankles or feet; (4) did not have any musculoskeletal, systemic and/or neurological disease; and (5) did not show any hypermobility or joint laxity sign. The protocol conformed to internationally accepted policy statements regarding the use of human participants in accordance with the Declaration of Helsinki Declaration and was approved by the Ovidius University of Constanta Nr. 126 din 18 March 2022. All participants gave their written informed consent to participate in the study after receiving a thorough explanation of the study’s protocol.

### 2.2. Instrumentation

Box Sit and Reach: The testing tool consists of a 31 cm high cube with a protrusion at the upper level along which the displacement in centimeters is determined (Figure 1). The external part, which constitutes the protrusion that extends beyond the front edge and therefore towards the tested participant, has a length of 23 cm, and in the center, there is the trolley through which its movement defines the elongation and therefore the various degrees in negative to positive values of flexibility [37].

Rubber ball: A 7 cm diameter rubber ball with a weight of 140 g (ATS, Arezzo, Italy), was used for the foot SMR (Figure 2). The rigidity of the ball does not allow any kind of deformation under foot and lower limb pressure.

### 2.3. Procedure and Data Collection

Testing was carried out in a Sport Performance Laboratory at a temperature of 20 °C and relative humidity of 51% according to previous studies [39,40]. The testing session was performed in the morning between 9.00 and 11.00 a.m. in order to avoid any kind of circadian influence on muscle flexibility [41]. The day before the testing session, in the same place and at the same time of the day, all the participants were familiarized with the S&R test. The results of this familiarization session were recorded.

During the test session, each participant was tested in four different moments across a one-hour testing experimental procedure (Figure 3):T0: corresponding to the first S&R test carried out on the participants;T1: corresponding to the second S&R test on the participants immediately after the SMR technique;T2: corresponding to the third test of the S&R, carried out 30 min after T1;T3: corresponding the fourth test of the S&R, carried out 60 min after T1.

S&R was chosen because of its unique ability to incorporate the lumbar spine and flexibility of the hamstring simultaneously while tensioning the posterior muscular chain [42]. The test was performed as reported in the original literature [37]: the participants sat on the floor barefoot, the legs were straight, the soles of the feet were placed flat against the box, forming an angle of 90 degrees with the ankles, and both knees were locked and pressed flat to the floor. From this position, the participants slowly stretched their hands forward with outstretched arms and pushed the measuring gage, reaching forward as far as possible, holding the maximal reach for 2 s [43] in order to allow the operator to read the result. Each participant performed each test (T0–T3) three times, and the average values were considered for the statistical analysis [44,45]. The S&R measures and the familiarization results were made by the same expert kinesiologist (having more than 10 years of experience).

To be consistent with previous works on SMR and S&R [29,30,31,46,47], no warming-up procedures were performed before the S&R test.

According to the results obtained in T0, three sub-groups were created considering the level of posterior muscular chain flexibility. Specifically, a flexible group (FG), a group with average flexibility (AG), and a stiff group (SG–11 participants) were created. The groups’ categorization was based on flexibility level at T0. In fact, the FG, the AG, and the SG showed S&R values of > 10 cm, > 0 but < 10 cm and < 0 cm respectively [48]. The FG counted 10 participants (4 males and 6 females), AG had 15 participants (4 males and 11 females), and SG had 11 participants (9 males and 2 females). Because of the total number of participants, the three sub-groups were mixed for gender. Previously published data show no influence of gender on the SMR acute improvements for S&R performance [47].

### 2.4. Intervention

Immediately after the measurement of the posterior muscular chain flexibility, each individual performed a session of SMR of the plantar fascia of both feet. Participants performed the SMR training in an upright position, holding their hands on the wall to keep the balance and to allow the application of consistent pressure on the rubber ball, avoiding too much or too low-pressure application. The lead researcher (and expert kinesiologist) instructed participants regarding the SMR procedure. The rubber ball was placed under the sole of the foot, precisely on the plantar arch. The kinesiologist instructed each participant to move slowly the foot, seeking a kind of painful mass or nodule perceived within the same muscle band, called a trigger point. Once the trigger point had been self-identified by the participant, pressure was applied in order to feel slight pain and temporary discomfort equivalent to a pain level of 7 out of 10 [16,18]. Three bouts of myofascial release massage were performed for each foot, each bout lasted 30 s. The kinesiologist randomly assigned the starting foot, and the work sequence was switching back and forth between the left and right foot. In the first bout, the trigger point was identified and pressed. In the second bout, slow circling movements were performed around the trigger point. In the third and last bout, pressure was still applied to the trigger point combining flexion and extension movements of the toes. The intervention had a total duration of 3 min.

All the participants completed the intervention, and no dropout cases or adverse effects of the treatment were registered in this study. Immediately after the intervention, each participant performed the T1 S&R test. Then, a rest period of 30 min was observed until T2 and again 30 min until the final S&R test at T3. During the rest time, the participants were free to walk around the laboratory or to take a sit. Any other type of activity was forbidden (e.g., jumping, squatting).

### 2.5. Statistical Analysis

Descriptive data were reported as mean and standard deviation (SD). The Shapiro–Wilk test was applied to check that data were normally distributed. Data were analyzed using a repeated-measures analysis of variance (ANOVA) with Fisher values and Bonferroni post hoc corrections to look for differences in the S&R values across the four measurements taken over 60 min. The effect size was also calculated (eta squared, η^2^) for a better interpretation of the results (values of 0.01, 0.06, and above 0.15 were considered small, medium, and large, respectively). Repeated ANOVA measures were undertaken for the sub-groups separately, while a Kruskal–Wallis test was used to look for differences in the flexibility at T0 for the different subgroups. A significance level of α = 0.05 was adopted and all data were analyzed with Statistical Package for the Social Science (SPSS), version 27.0 (SPSS Inc., Chicago, IL, USA).

## 3. Results

Data analysis showed an effect of the SMR on the plantar surface of both feet on the S&R results over time (Table 1).

No differences were present for the entire sample between the S&R values measured in familiarization and test session at T0 (4.0 ± 8.5 and 4.3 ± 8.7 cm, respectively; *p* = 0.184).

For the whole sample (Figure 4), significant differences were found for the S&R between T0 and T1 (4.3 ± 8.7 and 5.7 ± 9.1 cm; *p* = 0.001), T0 and T2 (4.3 ± 8.7 and 6.4 ± 8.4 cm; *p* = 0.000), and T0 and T3 (4.3 ± 8.7 and 7.0 ± 8.7 cm; *p* = 0.000). Another significant difference was found for the S&R between T1 and T3 (5.7 ± 9.1 and 7.0 ± 8.7 cm; *p* = 0.002).

A primary stratification of the entire sample was made by dividing by sex. The male group showed a lower level of flexibility with respect to the female group at the T0 (0.5 ± 9.3 cm and 7.7 ± 6.6 respectively; *p* = 0.036), T1 (1.4 ± 9.7 cm and 9.5 ± 6.5 respectively; *p* = 0.013), T2 (2.4 ± 8.8 cm and 10.1 ± 6.1 respectively; *p* = 0.010), and T3 (2.6 ± 9.4 cm and 11.0 ± 5.8 respectively; *p* = 0.009). The male group showed increasing flexibility over time, but no statistical differences were measured: 0.5 ± 9.3 cm at T0; 1.4 ± 9.7 cm at T1; 2.4 ± 8.8 cm at T2; 2.6 ± 9.4 cm at T3. For the female group, significant differences were found between T0 and T1 (7.7 ± 6.6 and 9.5 ± 6.5 cm; *p* = 0.000), T0 and T2 (7.7 ± 6.6 and 10.1 ± 6.1 cm; *p* = 0.004), and T0 and T3 (7.7 ± 6.6 and 11.0 ± 5.8 cm; *p* = 0.000). Another significant difference was found between T1 and T3 (9.5 ± 6.5 and 11.0 ± 5.8 cm; *p* = 0.016).

The entire sample was also divided into three subgroups, namely the FG, AG, and SG, according to the S&R values measured at T0. The three subgroups showed a significant difference for the S&R at T0 (*p* = 0.000). Obviously, the FG showed higher flexibility (13.7 ± 2.7 cm) than AG (5.9 ± 3.6 cm) and the same was for AG compared to SG (−6.5 ± 4.3 cm), respectively. The behavior of the subgroups over time was different after the SMR stimulation (Figure 5). The FG showed no significant increasing trend over time: 13.7 ± 2.7 cm at T0; 14.6 ± 3.4 cm at T1; 14.5 ± 4.4 cm at T2; and 15.5 ± 4.1 cm at T3. The AG showed a behavior very similar to the whole sample, with T0 being significantly lower than T1, T2, and T3: 5.9 ± 3.6 cm at T0; 8.1 ± 4.1 cm at T1 (*p* = 0.001 compared to T0); 8.6 ± 4.3 cm at T2 (*p* = 0.003 compared to T0); and 9.0 ± 4.7 cm at T3 (*p* = 0.001 compared to T0). Finally, SG showed a significant difference between T0 and T2 and between T1 and T2: −6.5 ± 4.3 cm at T0; −5.8 ± 4.5 cm at T1 (*p* = 0.038 compared to T0); −3.8 ± 3.9 cm at T2 (*p* = 0.010 compared to T0); and −3.4 ± 4.6 cm at T3.

## 4. Discussion

With this work, we sought to determine the longer terms effects of the posterior muscular chain flexibility after practicing SMR on the plantar surface of both feet. This was undertaken by performing the S&R test immediately, 30 min, and 60 min after the SMR intervention. The SMR effects were investigated also for individuals categorized by different flexibility levels at baseline.

### 4.1. Posterior Muscular Chain Flexibility, Changes in the Whole Sample

The main result of this research is relative to the duration of the SMR along an hour time span and the entire sample showed a constant increase of the S&R values, with significant differences along the time. To date, this is the first study to measure the effect of SMR intervention on posterior muscular chain flexibility across one hour. The increasing flexibility immediately after the intervention (T1) is consistent with the literature [20,31,47], confirming the transmission of information along the myofascial chains [49,50,51]. However, the most important novelty of this research dwells in the prolonged duration of the effect, lasting for a full hour. This result allows sports science practitioners and coaches to use SMR procedures from a new perspective. In fact, the facilitation given by this intervention during training could accompany the practitioners for almost the entire training session, considering that a typical training session lasts 30–45 min or 60–120 min, respectively, for health or performance goals [52,53]. At the same time, it is fair to remember that during the one-hour test session, the participants did not perform physical activity, but they could only sit or walk around the laboratory, unlike during a training session. Despite the different conditions between the experiment and field activity, the results are very interesting from a practical point of view. Accordingly, it seems possible to perform SMR exercises under the foot before the training session, even in the locker room, and then go directly into the field. In fact, to give a practical use of these findings, it should be considered, for example, that less flexible athletes show more running economy [54,55], but at the same time, the running economy is related to tendon length [56]. Because SMR works on connective tissue and fascia, it could offer more flexibility without affecting the running performance [4,57]; although, some doubts remain about this relationship [58].

### 4.2. Posterior Muscular Chain Flexibility, Changes for Subgroups

Our results show that the female group was more flexible than the male, which is consistent with previous literature [59,60]. However, it is interesting to underline the behavior over time of these two groups. In fact, the female group showed a behavior identical to the entire sample, while no statistical differences were measured for the male group. A plausible explanation for the lack of statistical differences could be identified considering the higher data variation expressed by the male group. Accordingly, the coefficient of variation (CV) of the male group is 19.8, which is very high compared to the entire sample (CV 2.0) or with the other subgroups (CV 0.9, 0.1, 0.6, 0.2 for female, SG, AG, FG, respectively).

Despite the stratification of the sample based on sex, we also divided the entire data set based on the flexibility levels of the participants. The S&R test values at T0 were used to distinguish three subgroups: SG, AG, and FG. The effect of the SMR intervention on each group was different.

FG did not show significant changes over time. Accordingly, individuals with high flexibility levels seem to not be affected by an SMR stimulation of the muscular chain. This might happen as such an SMR intervention can be considered an under-threshold stimulus, and probably the short duration of the SMR is not sufficient to evoke any modification in this subgroup, which is like previous studies [29,30]. The question of the proper duration of the SMR, especially in a flexible cohort, is, however, still open [61], and more research is needed. Nevertheless, the characteristics of the SMR approach make it reasonable to speculate that such a procedure would not be indicated for more flexible individuals. In fact, it is well known that individuals that are more flexible answer differently to flexibility training with respect to rigid individuals [62].

AG and SG showed significant modifications over time, confirming the starting hypothesis, especially for SG. The behavior of AG and SG was totally different because AG showed a more constant increasing ramp, while SG showed a delayed effect of the SMR.

In the AG, it is possible to appreciate a strong acute effect protracting from T0 to T1 and then a stable increase of S&R values from T2 and T3. Therefore, in this group, the initial gain in terms of flexibility seems to be maintained constantly until T3.

In the SG, the increasing values of the S&R test can be considered delayed because differences in flexibility occur after 30 min from the SMR intervention. This is very interesting from a practical and sport science point of view because the SG seems to need more time to allow the myofascial chain adaptation to the new stimulus. Such a finding could depend on the higher density and stiffness of the connective tissue, which is a possible reason to explain the reduced posterior muscular chain flexibility [18,63]. In light of these results, one question is reasonable: could FG participants benefit from longer or more intense SMR sessions? Further research should investigate this aspect.

Previous similar studies [29,30,31,46,47] did not perform any kind of stratification of the entire sample, and therefore the approach of this study could be considered a novelty in the SMR research field. According to these results, trainers and sports coaches should carefully check the timing of the application of SMR intervention with their users and athletes, who must be identified based on their baseline flexibility levels. This could be the reason why other researchers avoided including participants with hypermobility [29].

### 4.3. Musculoskeletal and Fascial Aspects

The results of this research offer more information about the open discussion on fascial and muscular chains. How could it be possible that a plantar feet stimulation increases the S&R test performance by up to one hour? It is well-known that muscles are wrapped by connective tissue with different layers, the so-called fascial system. The fascia is body widespread, linking the skeletal muscles [49]. The fascial system has proprioceptive and nociceptive functions [64,65,66,67] and is innervated by mechanoreceptors [66]. When pressure or traction is applied, those may create a range of different responses that facilitate movement [29]. Because the plantar fascia is the most distal and caudal part of the posterior muscular chain [32], proper pressure on the plantar surface stimulates the mechanoreceptors of the fascia, allowing transitory facilitation of the whole chain [31], due to the viscoelastic properties of the fascial connective tissue.

### 4.4. Limitations

The lack of a control group could be seen as a limitation of this research, but according to the previous literature [29,31], it is well known that a difference exists between intervention and control groups. Therefore, the authors’ choice was to avoid splitting the whole sample into two smaller groups but to use the entire sample to increase the number of participants. This allowed us to create subgroups based on their flexibility level. Future research can be undertaken to add more information on the behavior of different flexibility groups by applying the same methods of this research but using a larger sample, considering at least a control and an intervention group with the same flexibility level (SG, AG, FG). In fact, the sample size should be considered another limitation of the present study.

In addition, the research is focused only on the flexibility of the entire posterior muscular chain over a one-hour span, but no data are available on the analytic flexibility modification (e.g., the flexibility of hamstring and calves) or ankle joint ROM. In fact, it could be very interesting to repeat the same protocol and to test analytically even the flexibility of plantar muscles, calves, hamstring, and lower back. This would help understand which part of the posterior muscular chain is more affected by the SMR intervention under the foot, and which part is more responsible for the flexibility improvement of the whole chain.

Finally, the last limitation of this research is the lack of information about the time needed to return to the flexibility baseline level. According to the results, after an hour from the intervention, the flexibility of the posterior muscular chain is still higher with respect to the baseline. Therefore, it could be interesting to understand how much time is needed to return to baseline values. Future research should investigate these aspects.

## 5. Conclusions

The acute effects of the feet plantar surface SMR on S&R performance last up to one hour after the intervention. The magnitude of the modification in posterior muscular chain flexibility depends on the individual flexibility level, as the SMR does not seem to produce effects in a flexible cohort, while it works satisfactorily in individuals with average flexibility and who were stiff. Trainers and sports professionals should consider and opportunely use the duration of the main effect of up to one hour in their training programs for health, fitness, and sports.

## Figures and Tables

**Figure 1 ijerph-20-00974-f001:**
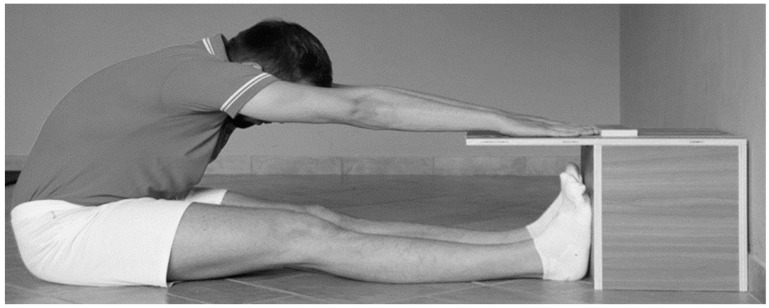
Sit and Reach test and measuring box.

**Figure 2 ijerph-20-00974-f002:**
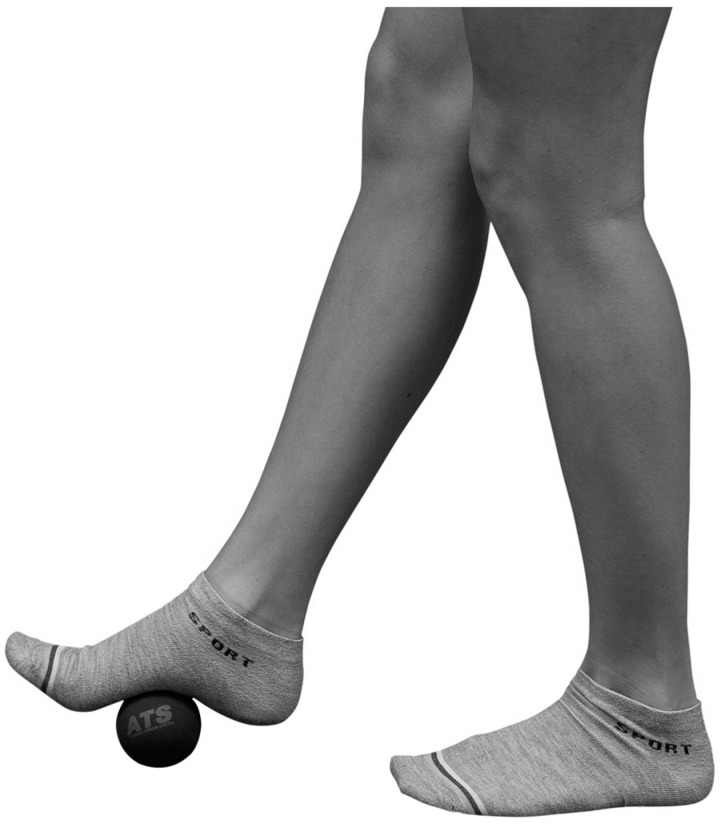
Rubber ball position under the foot for the Self Myofascial Release (SMR). Image modified by Russo et al. 2016 [38] with permission.

**Figure 3 ijerph-20-00974-f003:**
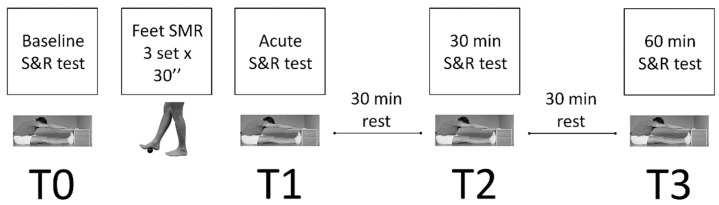
Experimental setting.

**Figure 4 ijerph-20-00974-f004:**
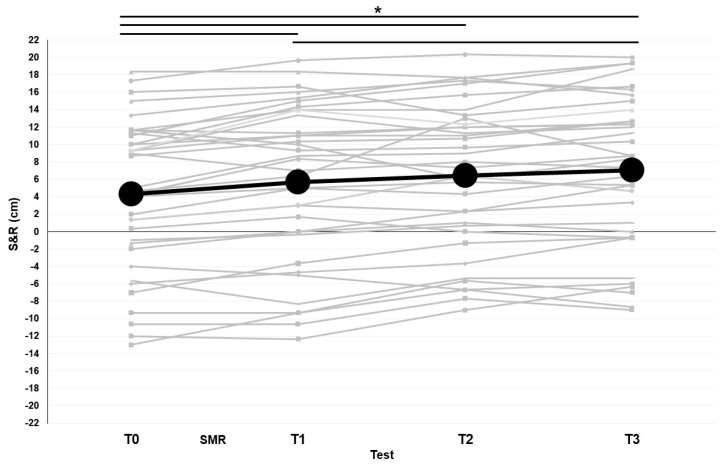
S&R values modifications over time before and after the SMR intervention. The black fat line represents the average values and thin grey lines are the real values of each participant. T0: test before intervention. T1: test after intervention. T2: test 30 min after intervention. T3: test 60 min after intervention. SMR: self-myofascial release. * Significant differences with *p* < 0.05.

**Figure 5 ijerph-20-00974-f005:**
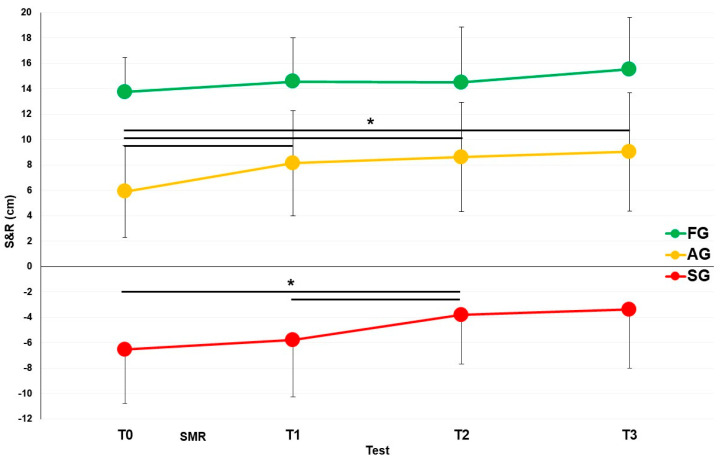
S&R modifications over the time before and after the SMR intervention in the three subgroups. SG: stiff group; AG: average group; FG: flexible group. T0: test before intervention. T1: test after intervention. T2: test 30 min after intervention. T3: test 60 min after intervention. SMR: self-myofascial release. * Significant differences with *p* < 0.05.

**Table 1 ijerph-20-00974-t001:** Results of ANOVA for repeated measures. Participants’ S&R values modification before and after SMR intervention.

	S&R (cm)						
Group (*n*)	T0	T1	T2	T3	F	η^2^	*p* Value
Entire sample (36)	4.3 ± 8.7	5.7 ± 9.1 *	6.4 ± 8.4 *	7.0 ± 8.7 *#	9.374	0.460	0.000
Males (17)	0.5 ± 9.3	1.4 ± 9.7	2.4 ± 8.8	2.6 ± 9.4	1.975	0.297	0.164
Females (19)	7.7 ± 6.6	9.5 ± 6.5 *	10.1 ± 6.1 *	11.0 ± 5.8 *#	10.457	0.662	0.000
FG (10)	13.7 ± 2.7	14.6 ± 3.4	14.5 ± 4.4	15.5 ± 4.1	1.198	0.339	0.378
AG (15)	5.9 ± 3.6	8.1 ± 4.1 *	8.6 ± 4.3 *	9.0 ± 4.7 *	8.838	0.688	0.002
SG (11)	−6.5 ± 4.3	−5.8 ± 4.5	−3.8 ± 3.9 *#	−3.4 ± 4.6	4.992	0.649	0.032

Note: data expressed as mean ± SD; T0 before intervention; T1 immediately after intervention; T2 30 min after intervention; T3 60 min after intervention; FG flexible group; AG average group; SG stiff group. Post-hoc comparison: * significant differences compared to T0; # significant difference compared to T1.

## Data Availability

The data that support the findings of this study are available from the corresponding author, upon reasonable request.

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
