# Peer review of "Self-Myofascial Release of the Foot Plantar Surface: The Effects of a Single Exercise Session on the Posterior Muscular Chain Flexibility after One Hour"

_ijerph, 2023, doi:10.3390/ijerph20020974_

Round 1
Reviewer 1 Report
· Introduction is correctly described. References are widely reported and the aim of the study in the final part is evidenced.
· Methodology is correctly presented. It could be interested to report also the setting description: air temperature and time of the day could influence sit and reach results.
· Results: correct the number of the figures (figure 3 appears 2 times instead of figures 4 and 5) at lines 198 and 215; correct significance at line 220 about “T1 compared to T3)
· Discussion is correctly presented and literature comparisons are deeply analyzed.
Many interesting questions are reported and several hypothesis are evidenced. Limits are deeply analyzed.
Author Response
Dear Editor and Reviewer,
We would like to thank you for the time allowed to this review process. As a result, we are submitting the revised version for a possible publication in this respectable Journal. Below, you can find our responses; each comment is followed by its respective reply. We made changes in the manuscript in order to address suggestions and make it clearer for the readers, we underlined in yellow the responses to your comments and we used the track changes to correct some misprint or to enhance some phrases of the manuscript. All authors have made sufficient contributions and have approved the submitted manuscript.
Sincerely,
The Authors
Legend:
R1(Reviewer 1)
A (Authors)
R1:
Introduction is correctly described. References are widely reported and the aim of the study in the final part is evidenced.
A:
Thank you very much; we are glad that you appreciated our introduction.
1) R1:
Methodology is correctly presented. It could be interested to report also the setting description: air temperature and time of the day could influence sit and reach results.
A:
Your suggestion is appreciated; we added the required information in lines 137-140.
2) R1:
Results: correct the number of the figures (figure 3 appears 2 times instead of figures 4 and 5) at lines 198 and 215; correct significance at line 220 about “T1 compared to T3)
A:
This part was revised according to your suggestions.
3) R1:
Discussion is correctly presented, and literature comparisons are deeply analyzed.
A:
Thank you very much; we are glad that you appreciated our discussion.
4) R1:
Many interesting questions are reported and several hypotheses are evidenced. Limits are deeply analyzed.
A:
Thank you very much for your constructive peer-review.
Reviewer 2 Report
Dear authors,
The manuscript it well write. Nevertheless, to perform some of the conclusions it seems that the methodological process must be improved.
If is not possible to add more data, please add more comparative studies in the discussion.
Also, I think that figure 3 must be deleted.
Author Response
Dear Editor and Reviewer,
We would like to thank you for the time allowed to this review process. As a result, we are submitting the revised version for a possible publication in this respectable Journal. Below, you can find our responses; each comment is followed by its respective reply. We made changes in the manuscript in order to address suggestions and make it clearer for the readers, we underlined in yellow the responses to your comments and we used the track changes to correct some misprint or to enhance some phrases of the manuscript. All authors have made sufficient contributions and have approved the submitted manuscript.
Sincerely,
The Authors
Legend:
R2 (Reviewer 2)
A (Authors)
1) R2:
The manuscript it well write. Nevertheless, to perform some of the conclusions it seems that the methodological process must be improved.
A:
Thank you very much for your suggestions. We improved the methodology section.
2) R2:
If is not possible to add more data, please add more comparative studies in the discussion.
A:
We added data dividing the sample by sex, lines 103-108, 239-248, 305-313. Moreover, we added an explanation in the discussion section with the data analysis
3) R2:
Also, I think that figure 3 must be deleted.
A:
Thank you very much. We think it could be useful to explain better the sequence of each phase of the study. We leave the decision about this choice to the editor. In any case we don’t have problem to remove the figure.
Reviewer 3 Report
The manuscript is designed to investigate the acute effects of a Self Myofascial Release (SMR) procedure applied to the feet' plantar muscles according to the participant’s flexibility level.
I have several restrictions regarding the arguments. The first is that SMR must be explained before indicating its use or indications. Indeed, informing the positive aspects of such a procedure is essential to provide the readers with a clear understanding of the purpose of the study.
Before considering the use of SMR, it is relevant for the authors to consider the magnitude of the gains. What is the mean or expected magnitude of the improvements? I need help before agreeing that SMR affects the posterior muscle chain.
I have serious concerns regarding the lack of evidence and robust data regarding using SMR (or any other approaches) to provide gains in segments the manipulated muscles do not spam. Depending on the improvements, it may be useful; otherwise, it may not!
Please consider rephrasing
“Moreover, there is no data available about the duration of the acute effects in different people with different levels of flexibility of the posterior muscular chain.”
“Moreover, no data is available regarding the acute duration of SMR in people with different posterior chain flexibility levels.”
Dividing by sex is interesting; however, considering the initial condition regarding ROM would be far more relevant and necessary. It is more difficult to impose changes in those with large ROM than those with small ROM.
Methodologically, using the sit-and-reach test is critical as it involves the back and shoulder grid muscles. The manipulation occurred at the foot's fascia, and the whole chain was tested. Therefore, there are too many intervening and uncontrolled factors. How stable were your S&R measurements? In addition, there is no indication of the time between T0 and T1. If there were at least one additional measure before T0, it would be relevant to reveal the testing effects. As it stands, the stability of your outcome measures has yet to be discovered. The lack of a control group is also critical. Note that every time the test is applied, your ROM increases. So, it is difficult to assume that such effects are related to the intervention rather than the test itself. Note that even after the application of the intervention once, gains are still occurring. Why?
The number of participants in each group must be disclosed. Intuitively, I may assume that 12 participants were allocated to each group, but the authors must confirm it. Thus, a sample size calculation is essential.
The comparisons between groups are useless as the authors used this specific criterion to separate the groups. I would be surprised if differences did not occur.
The limitation section requires revision and expansion.
The study announces that flexibility level would be considered. I see that no such analysis was considered. The flexibility level was not contemplated. I recommend revising the stats to fulfill the study's proposed aim. A regression equation would be far more elegant for such a purpose since your thresholds were arbitrarily defined. Additional analyses are required (e.g., minimal detectable changes – MDC) would add to a better approach.
Figures 2 & 3 could be merged if the authors use color to represent groups.
Most of the discussion is dedicated to reinforcing and repeating the findings, and little is discussed in terms of more relevant mechanisms for academic purposes. The debate regarding the groups is shallow and speculative.
Author Response
Dear Editor and Reviewer,
We would like to thank you for the time allowed to this review process. As a result, we are submitting the revised version for a possible publication in this respectable Journal. Below, you can find our responses; each comment is followed by its respective reply. We made changes in the manuscript in order to address suggestions and make it clearer for the readers, we underlined in yellow the responses to your comments and we used the track changes to correct some misprint or to enhance some phrases of the manuscript. All authors have made sufficient contributions and have approved the submitted manuscript.
Sincerely,
The Authors
Legend:
R3 (Reviewer 3)
A (Authors)
1) R3:
The manuscript is designed to investigate the acute effects of a Self Myofascial Release (SMR) procedure applied to the feet' plantar muscles according to the participant’s flexibility level.
I have several restrictions regarding the arguments. The first is that SMR must be explained before indicating its use or indications. Indeed, informing the positive aspects of such a procedure is essential to provide the readers with a clear understanding of the purpose of the study.
A:
Thank you, we appreciated your suggestion. We added the required information in lines 49-52.
2) R3:
Before considering the use of SMR, it is relevant for the authors to consider the magnitude of the gains. What is the mean or expected magnitude of the improvements? I need help before agreeing that SMR affects the posterior muscle chain.
A:
Thank you for your question. We understand your point of view and we tried to give some information about it. In our research, the mean magnitude of the improvement is more than 32% immediately after the intervention and 60% after one hour. Grieve et al. (https://pubmed.ncbi.nlm.nih.gov/26118527/) measured an improvement of 23% immediately after the SMR. Wilke et al. (https://pubmed.ncbi.nlm.nih.gov/30222474/) measured an improvement of 10% immediately after the SMR. These last two studies are the ones that inspired our research, pushing us to increase up to 60’ the follow up after intervention. In fact, we replicated a very similar protocol according to their data.
3) R3:
I have serious concerns regarding the lack of evidence and robust data regarding using SMR (or any other approaches) to provide gains in segments the manipulated muscles do not spam. Depending on the improvements, it may be useful; otherwise, it may not!
A:
Thank you for your comment. We cited several literature reviews on SMR in our paper (Beardsley et al., 2015; Kalichman et al., 2017; Behm et al. 2019; López-Torres et al., 2021; Ughreja et al., 2021; Ferreira et al., 2022) and none of them refers to a lack of evidence or usefulness about the improvement on flexibility and ROM after performing SMR procedures. Only one paper (Behm et al. 2019) questioned the use of the term “release” but not the final effects of the technique.
We began this study according to literature, because several reviews suggested increasing the research on this topic. In the light of this perspective, we welcome your comment, and we will be glad to continue our research to understand better the practical impact of the acute flexibility improvements on movement quality and performance. It is a very interesting suggestion for a next study starting from the present data.
4) R3:
Please consider rephrasing
“Moreover, there is no data available about the duration of the acute effects in different people with different levels of flexibility of the posterior muscular chain.”
“Moreover, no data is available regarding the acute duration of SMR in people with different posterior chain flexibility levels.”
A:
We added the data according to your suggestions in lines 82-84.
5) R3:
Dividing by sex is interesting; however, considering the initial condition regarding ROM would be far more relevant and necessary. It is more difficult to impose changes in those with large ROM than those with small ROM.
A:
Thank you for your comment. We added the data for each sex sub-group in lines 103-108, Table 1, lines 239-248 and lines 305-313.
6) R3:
Methodologically, using the sit-and-reach test is critical as it involves the back and shoulder grid muscles. The manipulation occurred at the foot's fascia, and the whole chain was tested. Therefore, there are too many intervening and uncontrolled factors. How stable were your S&R measurements? In addition, there is no indication of the time between T0 and T1. If there were at least one additional measure before T0, it would be relevant to reveal the testing effects. As it stands, the stability of your outcome measures has yet to be discovered. The lack of a control group is also critical. Note that every time the test is applied, your ROM increases. So, it is difficult to assume that such effects are related to the intervention rather than the test itself. Note that even after the application of the intervention once, gains are still occurring. Why?
A:
Thanks for your comment, we tried to answer with 3 points:
1) The day before the official test session, we made a familiarization session with the S&R and we measured the data; therefore, we added this information in lines 140-142 and lines 225-226. Moreover, a kinesiologist expert in this field performed each test session (lines 163-165).
2) T1 measures was performed immediately after the SMR intervention (line 193). The intervention had a total duration of 3 minutes; therefore, T0 and T1 were separated by the technical time to move from one side of the laboratory where the S&R was performed, to another side of the laboratory where the 3 minutes of intervention were administered and back again to the S&R side.
3) About the use of S&R and the residual effect of a single session of SMR (principal aim of our study), we used S&R test because we need to assess the posterior muscular chain and S&R is the most used and well-known test for this kind of evaluation. The test provides a reliable measure of the degree of flexibility of the posterior muscular chains (https://pubmed.ncbi.nlm.nih.gov/24570599/); moreover, as we wrote in the answer of comment 2, we aimed to resemble our study at other studies with a similar protocol and, these studies used S&R (lines 166-167 and 343-345). According to this point of view, we well know the limits of this test, at the same time the duration of the flexibility modification of the entire posterior muscular chain with a single intervention under the foot is the real novelty of this study. The 4.3 paragraph of the discussion gives more information about this point.
7) R3:
The number of participants in each group must be disclosed. Intuitively, I may assume that 12 participants were allocated to each group, but the authors must confirm it. Thus, a sample size calculation is essential.
A:
We added the data according to your suggestions in lines 103-108 and 172-174.
8) R3:
The comparisons between groups are useless as the authors used this specific criterion to separate the groups. I would be surprised if differences did not occur.
A:
Thanks you very much for your comment. We made the comparison at T0 just to underline that the used criterion has a real effectiveness in discriminating the flexibility groups.
9) R3:
The limitation section requires revision and expansion.
A:
We added more information according to your suggestions in lines 372-374, 379-383.
10) R3:
The study announces that flexibility level would be considered. I see that no such analysis was considered. The flexibility level was not contemplated. I recommend revising the stats to fulfill the study's proposed aim. A regression equation would be far more elegant for such a purpose since your thresholds were arbitrarily defined. Additional analyses are required (e.g., minimal detectable changes – MDC) would add to a better approach.
A:
We considered the “flexibility level” as the S&R value at T0 to balance the sub-group levels in baseline condition, according to the literature (lines 168-172).
11) R3:
Figures 2 & 3 could be merged if the authors use color to represent groups.
A:
Do you refer maybe to figure 4 and 5? We would be happy to maintain figure 4 because it allows us to show the behavior of each single participant. Whilst we appreciate your comment, we leave the decision about this to the Editor.
12) R3:
Most of the discussion is dedicated to reinforcing and repeating the findings, and little is discussed in terms of more relevant mechanisms for academic purposes. The debate regarding the groups is shallow and speculative.
A:
We added more information in the discussion, thank you.
Reviewer 4 Report
While the study is a simple study with limited application, the study is straightforward and easily repeatable for future studies to build off of these results. Within the abstract, line 26, the number of participants is given as 38, but 36, line 92, was given for the methods. As for the objectives, line 87, why was 1 hour selected for this study? The discussion within the introduction included limited research tracking benefits of SMR beyond immediate responses, but what lead up to 1 hour, instead of a longer timeframe? The authors do mention that potentially the 1 hour reflects a typical timeframe for a training session, but justification with research publications cited to support this conclusion would give a better understanding for the reason for this timeframe.
Within methods concerning the description of participants, line 95, authors mention "sport and fitness activity" of participants, but what was assessed specifically in quantifying in an objective manner their "activity" to determine that they were "physically active"? In line 149-150, participants are discussed concerning their division into sub groups in which at this point numbers per sub groups along with descriptions for the type of person within each sub group would be helpful. For example, while age, height, and weight is given for the whole group, this information needs to be given for each sub group along with number of males and females for each sub group.
In lines 159-160, the "trigger point" is mentioned, but how was that verified objectively that the "trigger point" was found for each participant and who was the identifier? Was it the participant, sports trainer, researcher? If it was someone besides the participant, was it the same person for each participant? Did a qualified individual work with the participant throughout the SMR and S&R test, and if so, what were their qualifications and did they work with all participants? For the process of the SMR, since both feet were worked, was the first foot utilized for the SMR randomized or was it the same foot, left or right, for each participant? Was foot selection for the SMR process done by the participant or by the authors? Did the SMR process include switching back and forth between left and right foot or was one side completed three times followed by the other side completed three times? Was this process consistent for all?
As for results, add to Table 1 the n values for each sub group. As for the discussion, lines 246-247, similar to what was mentioned in the abstract, the focus was on the results over the 1 hour time length would be similar to what a training session would include. However, the participants for this study only walked or sat between SMR sessions so that no other physical activity was done, unlike a training session. To come to the conclusion that SMR could last through a 1 hour training session, then, the authors should have had the participants doing a typical training session and complete S&R tests throughout the training and afterwards. In addition, keep in mind that training sessions can vary immensely not only in the time length, but the activities performed, thus, this is a broad conclusion for something that was not measured. Focus on the results specific to the activity recognizing the lack of a training session during the SMR within this study, and the fact that there is diversity in training activities and length within the industry.
Finally, within the discussion of limitations, sample size should be addressed. Although the total was 36, this number was divided into three groups. At this time, it is unsure if the number across the three sub groups is 12 participants or maybe even a group with a lower number. Nevertheless, in any case, 12 participants or lower is a small sample size for such a simple study. Without a more thorough evaluation of health status including clinical examination and diagnostic testing at the time of the study, the participants within the sub groups could have a more diverse health background impacting results.
Author Response
Dear Editor and Reviewer,
We would like to thank you for the time allowed to this review process. As a result, we are submitting the revised version for a possible publication in this respectable Journal. Below, you can find our responses; each comment is followed by its respective reply. We made changes in the manuscript in order to address suggestions and make it clearer for the readers, we underlined in yellow the responses to your comments and we used the track changes to correct some misprint or to enhance some phrases of the manuscript. All authors have made sufficient contributions and have approved the submitted manuscript.
Sincerely,
The Authors
Legend:
R4 (Reviewer 4)
A (Authors)
1) R4:
While the study is a simple study with limited application, the study is straightforward and easily repeatable for future studies to build off of these results.
A:
Thank you for your comment. Realizing a research that was repeatable was our goal.
2) R4:
Within the abstract, line 26, the number of participants is given as 38, but 36, line 92, was given for the methods.
A:
We confirmed 36 participants. Sorry it was a misprint, we updated this sentence.
3) R4:
As for the objectives, line 87, why was 1 hour selected for this study? The discussion within the introduction included limited research tracking benefits of SMR beyond immediate responses, but what lead up to 1 hour, instead of a longer timeframe? The authors do mention that potentially the 1 hour reflects a typical timeframe for a training session, but justification with research publications cited to support this conclusion would give a better understanding for the reason for this timeframe.
A:
Thank you for your comment; this point of view is it very relevant. We added the reference in lines 290-291.
We chose only one hour because, according to the previous literature, we expected to find a shorter duration of the acute effect and we wanted to show the return to the baseline values. It was surprising also for us to see that the SMR acute effect lasted up to one hour without a return to baseline S&R values. For this reason, we wrote in lines 384-388 that it could be interesting to know how much time is needed to return to baseline S&R performance.
4) R4:
Within methods concerning the description of participants, line 95, authors mention "sport and fitness activity" of participants, but what was assessed specifically in quantifying in an objective manner their "activity" to determine that they were "physically active"?
A:
We added more information according to your suggestions in lines 109-110.
5) R4:
In line 149-150, participants are discussed concerning their division into sub groups in which at this point numbers per sub groups along with descriptions for the type of person within each sub group would be helpful. For example, while age, height, and weight is given for the whole group, this information needs to be given for each sub group along with number of males and females for each sub group.
A:
Thank you for your comment. It was suggested even by another reviewer. In fact, we improved the description of the sample. You can find adjustments in lines 103-108 and 172-174.
6) R4:
In lines 159-160, the "trigger point" is mentioned, but how was that verified objectively that the "trigger point" was found for each participant and who was the identifier? Was it the participant, sports trainer, researcher? If it was someone besides the participant, was it the same person for each participant? Did a qualified individual work with the participant throughout the SMR and S&R test, and if so, what were their qualifications and did they work with all participants? For the process of the SMR, since both feet were worked, was the first foot utilized for the SMR randomized or was it the same foot, left or right, for each participant? Was foot selection for the SMR process done by the participant or by the authors? Did the SMR process include switching back and forth between left and right foot or was one side completed three times followed by the other side completed three times? Was this process consistent for all?
A:
We are glad to receive this comment. It gives us the opportunity to explain very well, with many details, the SMR intervention in order to allow other researcher to replicate the study. We added information according to your suggestions in lines 181-190.
7) R4:
As for results, add to Table 1 the n values for each sub group.
A:
We added “n” for each sub-group.
7) R4:
As for the discussion, lines 246-247, similar to what was mentioned in the abstract, the focus was on the results over the 1 hour time length would be similar to what a training session would include. However, the participants for this study only walked or sat between SMR sessions so that no other physical activity was done, unlike a training session. To come to the conclusion that SMR could last through a 1 hour training session, then, the authors should have had the participants doing a typical training session and complete S&R tests throughout the training and afterwards. In addition, keep in mind that training sessions can vary immensely not only in the time length, but the activities performed, thus, this is a broad conclusion for something that was not measured. Focus on the results specific to the activity recognizing the lack of a training session during the SMR within this study, and the fact that there is diversity in training activities and length within the industry.
A:
Thank you for this comment. It helped us improve our discussion. We added information according to your suggestions in lines 291-294.
8) R4:
Finally, within the discussion of limitations, sample size should be addressed. Although the total was 36, this number was divided into three groups. At this time, it is unsure if the number across the three sub groups is 12 participants or maybe even a group with a lower number. Nevertheless, in any case, 12 participants or lower is a small sample size for such a simple study. Without a more thorough evaluation of health status including clinical examination and diagnostic testing at the time of the study, the participants within the sub groups could have a more diverse health background impacting results.
A:
We updated according to your suggestions in line 372 and lines 373-374.
Round 2
Reviewer 3 Report
A definition must include what the underlying mechanisms are. Note That no clue is given regarding how it works! What evidence supports that (a) it modifies the manipulated tissue? (b) what it causes on the target tissue (is it mechanical, neural, or both)?
The authors mentioned that SMR is comparable to a self-induced massage. So, why a self-massage group was not used as a control? I understand the difficulties of using such an approach; however, the data must be presented. Note that concluding that no effect was produced (p=0.18) by treating all groups as one. Also, note that the SDs are considerably large (a CV of 200%) and do not allow one to understand the effect in each group. It is likely to have an impact when your groups are observed separately. The significant variation encourages analyzing groups individually. Otherwise, the whole analysis on this point is compromised.
Having your groups presenting a mixed profile (men and women) is a big problem. Note that your AG is predominantly formed by women (36% of males), while the SG is formed by almost three times more females than male participants. This is a nitid bias, especially when the “female was more flexible than male” (Line 305). I confess that I struggle to understand how the authors concluded that no statistical differences were found when one of the groups was formed by two female participants. I insist that this is a big issue, as you may have observed that your data (Table 1) indicated that no differences were found for the male participants (47% of your sample).
The request to present the data before and after applying the S&R test demand that the authors include a column in your tables and a time series point in your figures. I understand it replicates previous studies; however, remember that they claim this issue is a limitation that must be tackled (which rolls from study to study with no specific approach).
Sample size calculations are not presented on the lines indicated in the response. In other words, what is the number of participants given the power and significance? (Please visit https://sciencing.com/meaning-sample-size-5988804.html)
Author Response
Dear Reviewer 3,
We would like to thank you again for the time allowed to this review process. As in round 1, each comment is followed by its respective reply. We made some other changes in the manuscript in order to address your suggestions and make it clearer for the readers. All authors have made sufficient contributions and have approved the submitted manuscript.
Sincerely,
The Authors
Legend:
R3 (Reviewer 3)
A (Authors)
1) R3:
A definition must include what the underlying mechanisms are. Note That no clue is given regarding how it works! What evidence supports that (a) it modifies the manipulated tissue? (b) what it causes on the target tissue (is it mechanical, neural, or both)?
A:
We add some other well-known information according to literature in lines 55-63.
2) R3:
The authors mentioned that SMR is comparable to a self-induced massage. So, why a self-massage group was not used as a control? I understand the difficulties of using such an approach; however, the data must be presented.
A:
There is a misunderstanding in this point. We didn’t write in any part of the paper that SMR is “comparable” to a self-induced massage. We strictly followed the literature, in fact according to other authors (https://pubmed.ncbi.nlm.nih.gov/28532889/: https://pubmed.ncbi.nlm.nih.gov/26592233/; https://pubmed.ncbi.nlm.nih.gov/35010717/), we wrote the information that they gave to the scientific community: the SMR is a sub-category of the MFR. The MFR is manually performed by a therapist applying a massage to the patient’s body, therefore we wrote this sentence “to mobilize the targeted soft tissues with rolling devices, such as foam rollers or hard balls, which are used by applying a certain amount of pressure over the surface of the skin, and thus creating a self-induced massage effect.” It is the only phrase in the entire paper where we wrote “self-induced massage”. There is no comparison, as you wrote in the comment. Moreover the aim of our research is not to compare SMR with a similar manual self-induced massage; for this reason, there is no a control group working with this technique. It is a different research goal. Maybe it is useful for our next research but not consistent with the aim of the present research. We just wanted to measure the duration of the effect of a SMR technique that is used very often during training or in previous researches (https://pubmed.ncbi.nlm.nih.gov/34886078/; https://pubmed.ncbi.nlm.nih.gov/26118527/; https://pubmed.ncbi.nlm.nih.gov/34886078/; https://pubmed.ncbi.nlm.nih.gov/30222474/).
3) R3:
Note that concluding that no effect was produced (p=0.18) by treating all groups as one. Also, note that the SDs are considerably large (a CV of 200) and do not allow one to understand the effect in each group. It is likely to have an impact when your groups are observed separately. The significant variation encourages analyzing groups individually. Otherwise, the whole analysis on this point is compromised.
A:
Another misunderstanding is present in this point. The p value that you cite “(p = 0.18)” is referred to the difference between the familiarization test session (the day before the intervention) and the T0 of the test session. We add this information in lines 237-238 because in the peer-review round 1 you looked for information about the stability of our measures (you wrote us “If there were at least one additional measure before T0, it would be relevant to reveal the testing effects.”). We had the information and we added it. Actually, we explicit better that sentence, line 238.
The effect of SMR on the entire sample are well described in table 1.
We are glad to have cleared this aspect and the misunderstanding.
4) R3:
Having your groups presenting a mixed profile (men and women) is a big problem. Note that your AG is predominantly formed by women (36% of males), while the SG is formed by almost three times more females than male participants. This is a nitid bias, especially when the “female was more flexible than male” (Line 305). I confess that I struggle to understand how the authors concluded that no statistical differences were found when one of the groups was formed by two female participants. I insist that this is a big issue, as you may have observed that your data (Table 1) indicated that no differences were found for the male participants (47% of your sample).
A:
This comment has an error. The SG had 11 participants, 9 male and 2 females (line 184); in the comment, you wrote a different information and it is not correct. Nevertheless, the analysis for gender is present in the paper: females showed a significant effect of SMR but males did not, probably because their high inhomogeneity in S&R results. We can understand your point of view that gender could be a confounding variable but, at the same time, we have to follow the literature. Actually, the first research that studied “the occurrence of non-local exercise effects in a large sample” (https://pubmed.ncbi.nlm.nih.gov/30222474/) showed no influence of sex about the SMR acute improvements of S&R performance. Therefore, in the light of this data, we used the S&R baseline value of each participant to create the sub-groups FG, AG, SG. Previous published paper did not explicit the group composition: this paper https://pubmed.ncbi.nlm.nih.gov/26118527/ mixed males and females in a 12 participant group without declaring the males/females ratio and the same was done in this other paper https://pubmed.ncbi.nlm.nih.gov/34886078/. Conversely, we add and explicit this information in lines 182-186.
5) R3:
The request to present the data before and after applying the S&R test demand that the authors include a column in your tables and a time series point in your figures. I understand it replicates previous studies; however, remember that they claim this issue is a limitation that must be tackled (which rolls from study to study with no specific approach).
A:
We are not sure we understand the comment. We well described the process of the study, the SMR was performed between T0 and T1. Therefore, to be clearer, we add a text box “SMR” between T0 and T1 in the table, while it was already present in the figures.
6) R3:
Sample size calculations are not presented on the lines indicated in the response. In other words, what is the number of participants given the power and significance? (Please visit https://sciencing.com/meaning-sample-size-5988804.html)
A:
In lines 111-114 we declared the whole sample size calculation. During the analysis, after measuring the significant effect on the entire sample, the other sub-groups were created, according to their flexibility levels at T0. Nevertheless, using a sample size simulation for the same statistical test with an effect size F at 0.4, an error probability at 5% and the power of the analysis at 80% the recommended sample is 10 participants and we had that number. Moreover, previous data, published on this honorable journal (https://pubmed.ncbi.nlm.nih.gov/34886078/), used the same exercise on the same area of the foot with 16 people and without any kind of sample size calculation; the number of the participants is likely to us and the results are very stackable. Another published paper (https://pubmed.ncbi.nlm.nih.gov/26118527/ we cited in the references) used 12 participants with the same exercise and very stackable results, as well. Anyway, in lines 114-116 we add the declared information.